# ABCD2-I Score Predicts Unplanned Emergency Department Revisits within 72 Hours Due to Recurrent Acute Ischemic Stroke

**DOI:** 10.3390/diagnostics14111118

**Published:** 2024-05-28

**Authors:** Wei-Zhen Lu, Hui-An Lin, Sen-Kuang Hou, Sheng-Feng Lin

**Affiliations:** 1Department of Emergency Medicine, Taipei Medical University Shuang Ho Hospital, New Taipei 23561, Taiwan; weizhen123474@gmail.com; 2Department of Emergency Medicine, Taipei Medical University Hospital, Taipei 11031, Taiwan; sevenoking219@gmail.com (H.-A.L.); 992001@h.tmu.edu.tw (S.-K.H.); 3Department of Emergency Medicine, School of Medicine, College of Medicine, Taipei Medical University, Taipei 11031, Taiwan; 4Department of Public Health, School of Medicine, College of Medicine, Taipei Medical University, Taipei 11031, Taiwan; 5School of Public Health, College of Public Health, Taipei Medical University, 250 Wu-Hsing Street, Taipei 11031, Taiwan; 6Center of Evidenced-Based Medicine, Taipei Medical University Hospital, Taipei 11031, Taiwan

**Keywords:** acute ischemic stroke, carotid ultrasonography, transient ischemic attack, risk scoring model, emergency medicine

## Abstract

Background: the ABCD2 score is valuable for predicting early stroke recurrence after a transient ischemic attack (TIA), and Doppler ultrasound can aid in expediting stroke triage. The study aimed to investigate whether combining the ABCD2 score with carotid duplex results can enhance the identification of early acute ischemic stroke after TIA. Methods: we employed a retrospective cohort design for this study, enrolling patients diagnosed with TIA who were discharged from the emergency department (ED). The modified ABCD2-I (c50) score, which incorporates a Doppler ultrasound assessment of internal carotid artery stenosis > 50%, was used to evaluate the risk of acute ischemic stroke within 72 h. Patients were categorized into three risk groups: low risk (with ABCD2 and ABCD2-I scores = 0–4), moderate risk (ABCD2 score = 4–5 and ABCD2-I score = 5–7), and high risk (ABCD2 score = 6–7 and ABCD2-I score = 8–9). Results: between 1 January 2014, and 31 December 2019, 1124 patients with new neurological deficits were screened, with 151 TIA patients discharged from the ED and included in the analysis. Cox proportional hazards analysis showed that patients in the high-risk group, as per the ABCD2-I (c50) score, were significantly associated with revisiting the ED within 72 h due to acute ischemic stroke (HR: 3.12, 95% CI: 1.31–7.41, *p* = 0.0102), while the ABCD2 alone did not show significant association (HR: 1.12, 95% CI: 0.57–2.22, *p* = 0.7427). Conclusion: ABCD2-I (c50) scores effectively predict early acute ischemic stroke presentations to the ED within 72 h after TIA.

## 1. Introduction

In the emergency department (ED), non-contrast brain computed tomography (CT) serves as the standard diagnostic workup for acute ischemic stroke and transient ischemic attack (TIA) [1]. However, non-contrast brain CT alone exhibits low sensitivity for patients whose TIA has fully resolved [2]. In 2009, both the American Heart Association and American Stroke Association introduced a new definition of TIA, utilizing a “tissue-based” approach rather than a “time-based” approach [2,3]. Under this definition, any evidence of brain tissue infarction on a diffusion-weighted imaging (DWI) sequence of magnetic resonance imaging (MRI) in a patient—regardless of the duration of neurological deficits—is categorized as acute ischemic stroke [1].

Brain MRI not only accurately delineates the location and extent of tissue infarction, but also provides vascular imaging through magnetic resonance arteriography (MRA). Over the last decade, brain MRI has emerged as the preferred modality and, ideally, should be performed within 24 h of symptom onset [1]. However, obtaining immediate brain MRI is often challenging in many ED settings [1]. Consequently, diagnosing TIA based on non-contrast CT and clinical resolution of symptoms within 24 h is a reasonable approach in ED setting [4]. Despite many advantages, the tissue-based TIA definition is constrained by the availability of MRI [5,6,7] and may have limited applicability to population-based studies [8].

Evaluation of TIA necessitates vascular imaging. In 2013, a joint statement by the American College of Radiology, the American Society of Neuroradiology, and the Society of Neurointerventional Surgery advocated for neuroimaging to screen for carotid and vertebral artery stenosis in patients with TIA [9]. They recommended the use of non-contrast CT in combination with Doppler ultrasound or computed tomography angiography (CTA) for TIA patients when immediate brain MRI is unavailable [9]. CTA is generally safe and should not be delayed, even for patients with a history of chronic kidney disease [10]. However, the administration of iodinated contrast may pose concerns with respect to patients with hyperthyroidism and a history of allergic reaction. 

Duplex ultrasound for evaluating carotid artery stenosis offers advantages such as low cost, absence of contrast requirement, and minimal radiation exposure. A previous study [11] compared the diagnostic accuracy of Doppler ultrasound and intra-arterial angiography for carotid artery stenosis, demonstrating that the Doppler ultrasound had a sensitivity of 0.36 and specificity of 0.91 for detecting 50–69% of carotid artery stenosis and a sensitivity of 0.89 and moderate specificity of 0.84 for detecting 70–99% of carotid artery stenosis. 

In ED, the ABCD2 score [12,13] is the most widely used and simplest risk stratification tool for patients with TIA. However, a notable limitation of the ABCD2 score is its failure to incorporate the extent of carotid artery stenosis. In 2014, the ABCD3-I score or ABCD3-I (d,c/i) score [14,15] was introduced, encompassing additional items such as dual TIA, neuroimaging revealing old or new infarction on DWI sequence of MRI, and ipsilateral carotid artery stenosis or major cerebral artery > 50%. While the ABCD3-I score offers a more comprehensive assessment, it also adds complexity, expanding the total point range from 0–7 to 0–13. Addressing this concern, we propose a simplified modification to the ABCD2-I score by solely integrating the neuroimaging of Doppler ultrasound to evaluate carotid artery stenosis. In this study, our objective was to investigate whether the ABCD2-I score could effectively identify the high-risk group for recurrent acute ischemic stroke within 72 h among patients discharged from the ED.

## 2. Materials and Methods

### 2.1. Design and Study Population

This study employed a retrospective cohort design, which was applied to the ED of the Taipei Medical University Hospital, a level 1 advanced emergency response hospital in Taipei City. Approval for the study was obtained from the Joint Institutional Review Board of Taipei Medical University (reference identifier: N202103017). Informed consent was waived since de-linked and de-identified data were used.

Data extraction from our ED registry was performed using the International Classification of Diseases, Tenth Revision, Clinical Modification (ICD-10-CM) of I60-69 from 1 January 2014, to 31 December 2019, targeting patients with symptoms of transient neurological deficits. Subsequently, medical records were reviewed to select patients presenting to the ED with new neurological deficits. Exclusion criteria included patients admitted with acute ischemic stroke or hemorrhagic stroke, age < 20 years, lack of CT performance in the ED, no recent Doppler ultrasound within 6 months, and presence of stroke or TIA mimic diseases. The final cohort consisted of patients with transient neurological deficits discharged from the ED. 

Data collected included patient characteristics such as age, blood pressure at ED, clinical symptoms of slurred speech and unilateral weakness, medical history of hypertension, diabetes mellitus, acute ischemic stroke types, carotid ultrasonography, transcranial doppler, and brain imaging of CT or MRI. In Taiwan, ED revisits within 72 h represent an important medical quality indicator and, therefore, we followed the discharged patients with a tentative diagnosis of TIA for 72 h. Diagnosis of TIA utilized a “time-based” approach [4,16,17] and was based on resolution of neurological deficits within 24 h and no corresponding lesions on non-contrast CT. Acute ischemic stroke was defined as an episode of neurological dysfunction caused by focal cerebral infarction and a compatible evidence of infarction on diffusion-weighted imaging (DWI) sequences of magnetic resonance imaging [18]. Accordingly, acute ischemic stroke was classified into large vessel occlusion, small vessel occlusion, embolic infarction, and other subtypes according to the TOAST classification [18,19]. 

### 2.2. Sonographic Measurement

For carotid duplex analysis, morphological measurements with B-mode imaging and color flow were used to define low-grade stenosis (>30%). Moderate-grade stenosis (>50%; c50) and high-grade stenosis (>70%; c70) were determined based on morphological change and flow velocity. For grading internal carotid artery (ICA) stenosis, peak systolic velocities (PSVs) of >125 cm/s and >230 cm/s were used as the thresholds for moderate- and high-grade stenosis, respectively [20,21]. The resistance index (RI) was calculated using the formula (PSV—end-diastolic velocity)/PSV. A transcranial Doppler was performed to evaluate intracranial stenosis in patients with an accessible temporal bone window. For transcranial Doppler, intracranial stenosis > 50% was defined as PSV > 120 cm/s for vessels in anterior circulation and PSV > 140 cm/s for those in posterior circulation [22]. The ultrasound machine used for carotid vessel evaluation was Affiniti 70 with a linear probe (Koninklijke Philips N.V., Amsterdam, The Netherlands). Carotid duplex scans were performed by certified technicians and the results were reviewed and verified by physicians. Additionally, the method for the measurement of carotid stenosis adhered to the European Carotid Surgery Trial (ECST) criteria: the percentage of ICA stenosis = (1 − [diameter of the most stenotic part/estimated original diameter at the site of the stenosis]) × 100 [23].

### 2.3. ABCD2 Scoring Systems and Risk Group Stratification

Based on the ABCD2 scores [12,13], we modified the scoring systems of ABCD2-I by incorporating findings from the carotid duplex. The scoring details are summarized in Table 1. Four scoring systems were employed: ABCD2 score, ABCD2-I (c30), ABCD2-I (c50), and ABCD2-I (c70). All patients with TIA were classified into three risk groups based on their scores: (1) low risk (ABCD2 and ABCD2-I scores = 0–4), (2) moderate risk (ABCD2 score = 4–5 and ABCD2-I score = 5–7), and (3) high risk (ABCD2 score = 6–7 and ABCD2-I score = 8–9). The low-risk group was used as the reference group for risk comparison.

### 2.4. Statistical Analyses

Continuous and categorical variables were compared using the student’s *t* test and Pearson’s chi-square test (or Fisher’s exact test), respectively. For each risk group grade according to ABCD2 and ABCD2-I scores, the cumulative incidence for TIA patients was calculated using the Kaplan–Meier method, and the log-rank test was used to compare the survival curves. To assess the risk of subsequent acute ischemic stroke, the Cox proportional hazards model was employed to obtain the hazard ratio (HR) and 95% confidence interval (CI). The dependent variable was TIA patients revisiting within 72 h for acute ischemic stroke versus no revisit, with independent variables including each risk category of the ABCD2 and ABCD2-I scores and the covariates of age and sex. Subgroup analyses by acute ischemic stroke recurrence within 24 h, 24–48 h, and 48–72 h were performed. Characteristics on a continuous scale were tested by analysis of variance, and characteristics on a categorical scale were tested by Pearson’s chi-square test (or Fisher’s exact test). All statistical analyses were conducted using SAS 9.4 software (SAS Institute Inc., Cary, NC, USA). A two-sided *p* < 0.05 was considered to be statistically significant.

## 3. Results

### 3.1. Baseline Characteristics 

From 1 January 2014, to 31 December 2019, a total of 1124 patients were registered in our database. Among these, 151 patients presented with acute TIA and were discharged from their first visit to the ED (Figure 1). We categorized the patients into two groups: those who revisited the ED with acute ischemic stroke and those who did not. Sixty patients revisited the ED with acute ischemic stroke within 72 h. The characteristics of the enrolled patients are shown in Table 2. Among the 151 patients, the mean age was 68.2 ± 13.4 years, 42.4% of patients were women, 21.9% presented with slurred speech, and 50.3% presented with unilateral weakness. Hypertension was found in 67.6% of patients, while diabetes mellitus in 23.8%. There were no significant differences in baseline characteristics between the two groups. Most patients who revisited with acute ischemic stroke developed large vessel occlusion, as classified by the TOAST (trial of ORG 10172 in acute stroke treatment) criteria [19]. In the stroke group, systolic blood pressure was higher and slurred speech was less common. We further divided the case group into three subgroups on revisit time: within 24 h, 24–48 h, and 48–72 h. There was no significant difference in baseline characteristics among the subgroups (Table 3).

### 3.2. Sonographic Features 

The findings from carotid duplex analysis are shown in Table 2. Morphological measurements indicated a significantly higher proportion of low-grade stenosis in the common carotid artery (48.3% vs. 29.7%, *p* = 0.0202) and in the ICA (23.3% vs. 8.8%, *p* = 0.0132) among patients who revisited the ED within 72 h. Moreover, moderate stenosis of the ICA was significantly more common in the group that revisited the ED within 72 h (24.4% vs. 2.2%; *p* = 0.0071). Additionally, the RI of the common carotid artery (0.74 ± 0.07 vs. 0.71 ± 0.08, *p* = 0.0140), the RI of ICA (0.65 ± 0.09 vs. 0.61 ± 0.09, *p* = 0.0032), and the flow of the ICA (201.0 ± 70.4 vs. 232.6 ± 76.6 mL/min, *p* = 0.0104) showed significant differences between the two groups. There was no significance in intima thickness, total flow of the bilateral vertebral artery, or the proportion of severe ICA stenosis > 70%.

### 3.3. Risk Group Stratification Analyses

Risk group stratification analyses based on the ABCD2 and ABCD2-I score models are shown in Table 4. The mean score according to the ABCD2-I (c50) (mean: 5.13 ± 1.60 vs. 4.57 ± 1.56, *p* = 0.0335) showed significant differences between the two groups. The distribution across different risk groups also showed significant differences when using the ABCD2 and ABCD2-I (c50) models (*p* = 0.0397; Figure 2). In contrast, the ABCD2-I (c30) model did not effectively differentiate between the groups, as evidenced by a *p* value close to 0.05 (*p* = 0.0484). Additionally, the ABCD2 score (*p* = 0.1937) and ABCD2-I (c70) score (*p* = 0.3123) did not show significant differences in the proportions of the three risk levels between the groups.

### 3.4. Survival Analysis

The Kaplan–Meier analysis of each risk level in the revisit group is shown in Figure 3. To determine whether the three risk levels were associated with acute ischemic stroke, a Cox proportional hazards model analysis was performed (Table 5). Using the ABCD2-I (c50) scoring model, patients classified as high risk were significantly associated with a higher likelihood of revisiting the ED due to acute ischemic stroke (HR: 3.25, 95% CI: 1.39–7.62, *p* = 0.0066; adjusted HR: 3.12, 95% CI: 1.31–7.41, *p* = 0.0102). In contrast, high-risk patients according to the ABCD2 and ABCD2-I (c70) models were not significantly associated with subsequent acute ischemic stroke.

## 4. Discussion

Rapid triage of patients with TIA is crucial for emergency physicians and stroke neurologists when making decisions in the ED. This study investigated whether integrating carotid duplex into the ABCD2 score could aid in predicting TIA patients at risk of acute ischemic stroke within 72 h. Our findings suggest that employing the ABCD2-I (c50) score, a simplified modification to the ABCD2 score, is practical and effective. Patients classified into the high-risk group (total points of 8–9) using the ABCD2-I (c50) model had a three-fold increased risk of revisiting the ED within 72 h compared to those classified as low risk.

The majority of those patients with TIA and stroke recurrence within 72 h were found to have large vessel occlusion, followed by small vessel occlusion and embolic infarction. Large vessel occlusion is known to have the highest rates of early stroke recurrence [24,25,26], and our study supports this, showing a high prevalence (46.7%) of revisit patients with large vessel occlusion (Table 2). The existing literature also identifies carotid artery stenosis as a major cause of TIA [24,27]. While incorporating sonographic findings into the ABCD2 score enhances the identification of asymptomatic large vessel occlusion, it does not imply that the scoring system is inappropriate for small vessel occlusion. Our cohort included patients (30%) with small vessel occlusion who revisited the ED.

A distinctive feature of our study was its ED settings. Carotid ultrasonography is suitable in ED settings due to its noninvasive nature, ease of use, and ability to provide real-time vascular imaging. When incorporating the findings of carotid duplex into the ABCD2 score, ICA stenosis > 50% was the most effective parameter for identifying patients who would develop acute ischemic stroke within 72 h (Table 4). In the moderate-risk group (ABCD2-I (c50) scores of 5–7), 50.0% revisited with acute ischemic stroke and 48.4% were discharged without a stroke. In the high-risk group (scores of 8–9), 11.7% revisited and 2.2% were discharged. These findings indicate that patients with an ABCD2-I (c50) score ≥ 5 should be closely monitored to prevent the occurrence of acute ischemic stroke.

Large vessel occlusion is known to have the highest rate of early stroke recurrence [24,25,26], and our analysis supports this, showing a high prevalence (46.7%) of revisit patients with large vessel occlusion (Table 2). Studies identify carotid artery stenosis as a major cause of TIA [24,27]. The methods for carotid artery intervention include carotid endarterectomy and carotid artery stenting. According to the “2021 Guideline for the Prevention of Stroke in Patients With Stroke and TIA”, published by the American Stroke Association [28], patients with a TIA and symptomatic moderate-grade (50–69%) or high-grade (70–99%) extracranial large vessel occlusion, documented via invasive or noninvasive imaging modalities, are advised to be managed with carotid endarterectomy or carotid artery stenting (Class of Recommendation 1). Despite not being explicitly documented in the guidelines, studies [29,30,31] recommend that carotid intervention shall be performed within 2 weeks. Excessive delay is believed to cause numerous subsequent acute ischemic stroke events [29]. However, the 2017 Carotid Alarm Study [32] found higher complication rates of death and/or any stroke for the symptomatic carotid stenosis within 48 h. Consequently, carotid intervention is considered safe when performed between 3 days and 7 days.

The ABCD3 score [33] includes an item called “dual TIA in preceding 7 days”, which we did not add to our models for several reasons. Our study tracked patients for three days post-visit, making “dual TIA” unsuitable. In addition, the ABCD3 score was developed in secondary care settings and designed for a longer follow-up time (30–90 days) [14,15,33]. Many TIA patients are older and may struggle to describe their symptoms accurately [34], complicating the inclusion of “dual TIA”. Our goal was to create a simple, practical scoring model for the ED. Therefore, ABCD2 or ABCD2-I are more suitable for ED settings. 

Our population had a high prevalence of hypertension and diabetes mellitus, consistent with a national survey data [35]. During the study period, from 2014 to 2019, we adhered to the 2015 Guidelines of the Taiwan Society of Cardiology and the Taiwanese Hypertension Society [36]. For patients with a history of TIA or acute ischemic stroke, antihypertensive medications were prescribed if the blood pressure exceeded 220/120 mmHg during the initial 24 h of the acute stage. Thereafter, blood pressure management aimed for a target blood pressure of <140/90 mmHg. For diabetes mellitus, we followed the Taiwan guidelines for the general management of patients with acute ischemic stroke [37] and relevant studies [38,39]. Blood glucose was controlled and maintained within the range of 140 mg/dl to 180 mg/dl.

This study has some limitations. First, this study was conducted in a single center, which may limit the generalizability of the findings. The short follow-up period and retrospective design may have introduced unmeasurable confounding effects. However, our prevalence of stroke subtypes and the proportions of moderate and high grades of internal carotid stenosis (>50%) were similar to the national surveys conducted in Asia [14,33,40,41,42,43,44]. This consistency suggests that there was no pre-selection in our cohort. Second, due to the brief duration of treatment with single or dual antiplatelet or anticoagulant therapy, we cannot infer the protective effect of antiplatelets or their association with early ED revisits due to acute ischemic stroke. Third, the limited number of high-grade ICA stenosis >70% cases in our study might have resulted in insufficient statistical power. Fourth, since our study focused on the rapid assessment of the ABCD2 score as well as ICA stenosis percentage in emergency settings, we did not record plaque vulnerability in this study. Further prospective studies with larger sample sizes are needed to validate our results. Lastly, while our study could not determine an intervention method to prevent acute ischemic stroke after TIA, dual antiplatelet therapy may afford a potential solution. Evidence from the Clopidogrel with Aspirin in Acute Minor Stroke or Transient Ischemic Attack (CHANCE) trial suggests that dual antiplatelet therapy with clopidogrel and aspirin is more effective than aspirin alone for the prevention of acute ischemic stroke over 90 days [45] and 1 year [46]. This evidence could support the consideration of dual antiplatelet therapy for patients identified as being at high risk by the ABCD2-I score.

## 5. Conclusions

In conclusion, our study demonstrates that the ABCD2-I (c50) score is more effective in predicting an early revisit to the ED within 72 h due to an acute ischemic stroke. The duplex finding of ICA stenosis >50% in a patient with TIA serves as a red flag sign of early acute ischemic stroke. Patients with TIA classified as moderate and high risk according to the ABCD2-I (c50) score should be closely monitored when discharged from the ED.

## Figures and Tables

**Figure 1 diagnostics-14-01118-f001:**
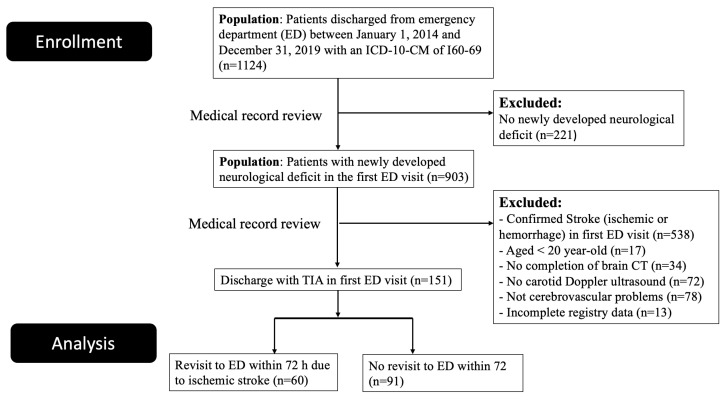
Flow diagram of patient enrollment.

**Figure 2 diagnostics-14-01118-f002:**
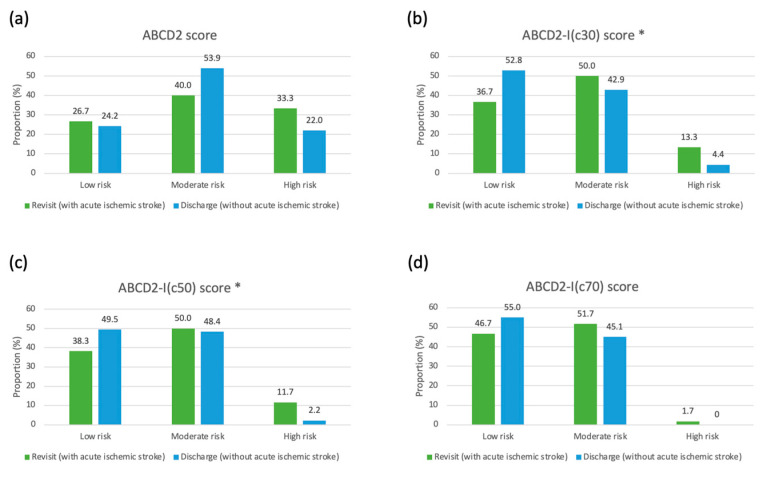
Proportions of low-, moderate-, and high-risk levels in patients with TIA who did or did not revisit the ED within 72 h after acute ischemic stroke. (**a**) ABCD2, (**b**) ABCD2-I (c30), (**c**) ABCD2-I (c50), and (**d**) ABCD2-I (c70) scores. Abbreviations: ED; emergency department, TIA; transient ischemic attack. * Significance was defined as *p* < 0.05 for Pearson’s chi-square test.

**Figure 3 diagnostics-14-01118-f003:**
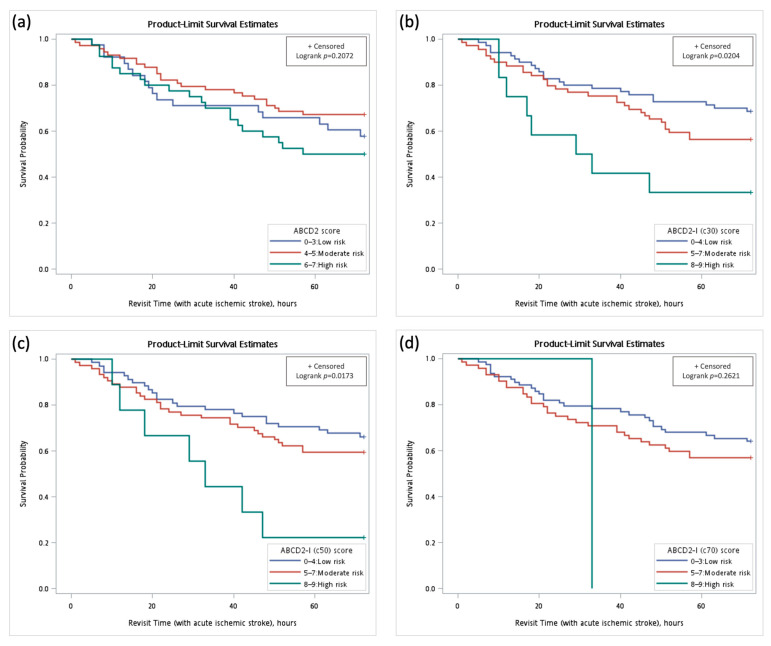
Kaplan–Meier curves of low-, moderate-, and high-risk levels in patients with TIA revisiting the ED within 72 h after acute ischemic stroke. (**a**) ABCD2, (**b**) ABCD2-I (c30), (**c**) ABCD2-I (c50), and (**d**) ABCD2-I (c70) scores. Abbreviations: ED; emergency department, TIA; transient ischemic attack.

**Table 1 diagnostics-14-01118-t001:** Scoring systems of ABCD2, ABCD2-I (c30), ABCD2-I (c50), and ABCD2-I (c70).

Scoring Systems	ABCD2	ABCD2-I (c30)	ABCD2-I (c50)	ABCD2-I (c70)
Age ≥ 60 y	1	1	1	1
BP ≥ 140/90 mmHg	1	1	1	1
Clinical symptoms				
Slurred speech without weakness	1	1	1	1
Unilateral weakness	2	2	2	2
Duration of symptoms				
10–59 min	1	1	1	1
≥60 min	2	2	2	2
Diabetes mellitus	1	1	1	1
Carotid duplex				
Any side of ICA stenosis > 30%	NA	2	NA	NA
Any side of ICA stenosis > 50%	NA	NA	2	NA
Any side of ICA stenosis > 70%	NA	NA	NA	2
Total points (maximum)	7	9	9	9

Abbreviations: BP, blood pressure; c30, internal carotid stenosis > 30%; c50, internal carotid stenosis > 50%; c70, internal carotid stenosis > 70%; ICA, internal carotid artery; NA, not-applicable.

**Table 2 diagnostics-14-01118-t002:** Characteristics of transient ischemic attack patients who did and did not revisit the emergency department within 72 h.

Characteristic Variables	Total	Revisit (with Acute Ischemic Stroke)	Discharge (without Acute Ischemic Stroke)	*p* Value
	(*n* = 151)	(*n* = 60)	(*n* = 91)	
Age	68.2 ± 13.4	69.8 ± 12.4	67.2 ± 14.0	0.2488
Female sex (%)	42.4% (64/151)	40.0% (24/60)	44.0% (40/91)	0.6302
First visit				
Systolic BP	157.3 ± 28.3	164.4 ± 31.9	152.6 ± 24.6	0.0174 *
Diastolic BP	87.1 ± 16.6	89.3 ± 18.4	85.7 ± 15.3	0.1941
ED revisit				
Systolic BP	-	155.7 ± 27.8	-	-
Diastolic BP	-	88.1 ± 19.7	-	-
Symptoms and comorbidity				
Slurred speech (%)	21.9% (33/151)	13.3% (8/60)	27.5% (25/91)	0.0397 *
Unilateral weakness (%)	50.3% (76/151)	43.3% (26/60)	55.0% (50/91)	0.1626
Hypertension (%)	67.6% (102/151)	76.7% (46/60)	61.5% (56/91)	0.0520
Diabetes (%)	23.8% (36/151)	26.7% (16/60)	22.0% (20/91)	0.5082
Ischemic stroke classification				
Large artery (%)	-	46.7% (28/60)	-	-
Small artery (%)	-	30.0% (18/60)	-	-
Embolic stroke (%)	-	3.3% (2/60)	-	-
Undetermined (%)	-	20.0% (12/60)	-	-
Sonographic findings				
Intima thickness	0.90 ± 0.31	0.89 ± 0.26	0.91 ± 0.35	0.7870
Intima thickness > 1 mm	18.5% (28/151)	20.0% (12/60)	17.6% (16/91)	0.7084
CCA stenosis > 30%	37.1% (56/151)	48.3% (29/60)	29.7% (27/91)	0.0202 *
CCA stenosis > 50%	17.9% (27/151)	25.0% (15/60)	13.2% (12/91)	0.0638
CCA stenosis > 70%	2.7% (4/151)	2.2% (2/60)	3.3% (2/91)	0.6707
ICA stenosis > 30%	14.6% (22/151)	23.3% (14/60)	8.8% (8/91)	0.0132 *
ICA stenosis > 50%	6.6% (10/151)	24.2% (8/60)	2.2% (2/91)	0.0071 *
ICA stenosis > 70%	0.7% (1/151)	1.7% (1/60)	0% (0/91)	0.3974
CCA PSV (cm/s)	70.2 ± 29.0	68.8 ± 38.8	71.3 ± 18.3	0.6200
CCA RI	0.72 ± 0.08	0.74 ± 0.07	0.71 ± 0.08	0.0140 *
CCA flow (mL/min)	380.9 ± 105.3	362.6 ± 111.0	394.9 ± 99.1	0.0604
ICA PSV (cm/s)	62.2 ± 26.7	64.8 ± 35.6	60.3 ± 17.5	0.3448
ICA RI	0.63 ± 0.09	0.65 ± 0.09	0.61 ± 0.09	0.0032 *
ICA Flow (mL/min)	219.0 ± 75.4	201.0 ± 70.4	232.6 ± 76.6	0.0104 *
Bilateral VA flow	129.2 ± 48.6	124.5 ± 42.1	132.6 ± 53.1	0.4789
Bilateral VA flow < 100 mL/min	65.6% (99/151)	61.7% (37/60)	68.1% (62/91)	0.8340
Intracranial vessel stenosis > 50%	26.2% (17/65)	29.0% (9/31)	23.5% (8/34)	0.6141

BP, blood pressure; CCA, common carotid artery; ED, emergency department; ICA, internal carotid artery; PSV, peak systolic velocity; RI, resistance index; VA, vertebral artery. * Significance was defined as *p* < 0.05.

**Table 3 diagnostics-14-01118-t003:** Characteristics of patients with new-onset acute ischemic stroke who revisited the emergency department (*n* = 60).

	Day 1	Day 2	Day 3	*p* Value
Number	32 (53.3%)	20 (33.3%)	8 (13.3%)	
Age	67.8 ± 13.8	72.2 ± 11.3	71.6 ± 8.0	0.4239
Female sex (%)	34.4% (11/32)	50.0% (10/20)	37.5% (3/8)	0.5753
First visit				
Systolic BP	166.3 ± 27.7	168.9 ± 39.1	143.6 ± 24.6	0.1831
Diastolic BP	91.2 ± 18.4	88.6 ± 19.8	85.4 ± 15.7	0.7269
ED revisit				
Systolic BP	158.7 ± 30.8	153.5 ± 27.3	147.2 ± 14.8	0.6147
Diastolic BP	92.7 ± 19.5	85.4 ± 19.3	79.8 ± 16.9	0.2243
Symptoms and comorbidity				
Slurred speech (%)	12.9% (4/31)	20.0% (4/20)	0 (0/8)	0.5127
Unilateral weakness (%)	45.2% (6/31)	45.0% (7/20)	37.5% (3/8)	0.9221
Hypertension (%)	71.0% (22/31)	80.0% (16/20)	87.5% (7/8)	0.5508
Diabetes (%)	19.4% (6/31)	35.0% (7/20)	37.5% (3/5)	0.3894
Stroke classification				0.8172
Large artery (%)	46.9% (15/32)	40.0% (8/20)	62.5% (5/8)	
Small artery (%)	25.0% (8/32)	40.0% (8/20)	25.0% (2/8)	
Embolic stroke (%)	3.1% (1/32)	5.0% (1/20)	0% (0/8)	
Undetermined (%)	25.0% (8/32)	15.0% (3/20)	12.5% (1/8)	
Point score				
ABCD2 score	4.4 ± 1.2	4.9 ± 1.3	4.5 ± 1.4	0.4140
ABCD2-I (c30)	4.9 ± 1.7	5.4 ± 1.5	4.8 ± 1.5	0.5175
ABCD2-I (c50)	4.7 ± 1.6	5.2 ± 1.5	4.8 ± 1.5	0.4876
ABCD2-I (c70)	4.4 ± 1.2	5.0 ± 1.4	4.5 ± 1.4	0.3214

BP, blood pressure; ED, emergency department.

**Table 4 diagnostics-14-01118-t004:** Predicting scores of transient ischemic attack patients with and without an emergency department revisit within 72 h.

(*n* = 151)	Total	Revisit (with Acute Ischemic Stroke)	Discharge (without Acute Ischemic Stroke)	*p* Value
Number	151	60 (39.7%)	91 (60.3%)	
Point score				
ABCD2 score, mean	4.45 ± 1.34	4.60 ± 1.26	4.35 ± 1.39	0.2665
ABCD2 score, distribution				0.1937
0–3: Low risk (%)	25.2% (38/151)	26.7% (16/60)	24.2% (22/91)	
4–5: Moderate risk (%)	48.3% (73/151)	40.0% (24/60)	53.9% (49/91)	
6–7: High risk (%)	26.5% (40/151)	33.3% (20/60)	22.0% (20/91)	
ABCD2-I (c30) score, mean	4.74 ± 1.63	5.07 ± 1.62	4.53 ± 1.60	0.0450 *
ABCD2-I (c30) score, distribution				0.0484 *
0–4: Low risk (%)	46.4% (70/151)	36.7% (22/60)	52.8% (48/91)	
5–7: Moderate risk (%)	45.7% (69/151)	50.0% (30/60)	42.9% (39/91)	
8–9: High risk (%)	8.0% (12/151)	13.3% (8/60)	4.4% (4/91)	
ABCD2-I (c50) score, mean	4.79 ± 1.59	5.13 ± 1.60	4.57 ± 1.56	0.0335 *
ABCD2-I (c50) score, distribution				0.0397 *
0–4: Low risk (%)	45.0% (68/151)	38.3% (23/60)	49.5% (45/91)	
5–7: Moderate risk (%)	49.0% (74/151)	50.0% (30/60)	48.4% (44/91)	
8–9: High risk (%)	6.0% (9/151)	11.7% (7/60)	2.2% (2/91)	
ABCD2-I (c70) score, mean	4.6 ± 1.5	4.9 ± 1.5	4.4 ± 1.5	0.0569
ABCD2-I (c70) score, distribution				0.3123
0–4: Low risk (%)	51.7% (78/151)	46.7% (28/60)	55.0% (50/91)	
5–7: Moderate risk (%)	47.7% (72/151)	51.7% (31/60)	45.1% (41/91)	
8–9: High risk (%)	0.7% (1/151)	1.7% (1/60)	0 (0/91)	

c30, internal carotid stenosis > 30%; c50, internal carotid stenosis > 50%; c70, internal carotid stenosis > 70%. * Significance was defined as *p* < 0.05.

**Table 5 diagnostics-14-01118-t005:** Hazard ratios for transient ischemic attack patients revisiting with ischemic stroke in 72 h.

(*n* = 151)	HR (95% CI)	*p* Value	Adjusted HR	*p* Value
Model 1 of ABCD2 score				
ABCD2 score, low risk	Reference	-		
ABCD2 score, moderate risk	0.74 (0.39–1.39)	0.3498	0.72 (0.38–1.36)	0.3081
ABCD2 score, high risk	1.21 (0.63–2.34)	0.5675	1.12 (0.57–2.22)	0.7427
Model 2 of ABCD2-I (c30) score				
ABCD2-I (c30), low risk	Reference			
ABCD2-I (c30), moderate risk	1.49 (0.86–2.58)	0.1578	1.46 (0.83–2.55)	0.1855
ABCD2-I (c30), high risk	3.02 (1.34–6.80)	0.0077 *	2.79 (1.20–6.46)	0.0170 *
Model 3 of ABCD2-I (c50) score				
ABCD2-I (c50) score, low risk	Reference			
ABCD2-I (c50), moderate risk	1.28 (0.74–2.20)	0.3770	1.24 (0.72–2.15)	0.4364
ABCD2-I (c50), high risk	3.25 (1.39–7.62)	0.0066 *	3.12 (1.31–7.41)	0.0102 *
Model 4 of ABCD2-I (c70) score				
ABCD2-I (c70), low risk	Reference			
ABCD2-I (c70), moderate risk	1.30 (0.78–2.16)	0.3208	1.26 (0.74–2.12)	0.3923
ABCD2-I (c70), high risk	3.91 (0.53–29.02)	0.1825	3.64 (0.49–27.23)	0.2084

c30, internal carotid stenosis > 30%; c50, internal carotid stenosis > 50%; c70, internal carotid stenosis > 70%; HR, hazard ratio. All models were adjusted for age and sex. * Significance was defined as *p* < 0.05.

## Data Availability

Data is unavailable due to privacy or ethical restrictions according to the Joint Institutional Review Board of Taipei Medical University.

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
