# Peer review of "ABCD2-I Score Predicts Unplanned Emergency Department Revisits within 72 Hours Due to Recurrent Acute Ischemic Stroke"

_diagnostics, 2024, doi:10.3390/diagnostics14111118_

Round 1

Reviewer 1 Report

Comments and Suggestions for Authors

Although it is potentially a valuable paper, there are several problems that should be addressed before proper review.

1.  In general population a majority of strokes are of cardiac origin (usually due to atrial fibrillation-associated embolism; here all patients presented with stenoses of their carotid/cerebral arteries. Probably there was a pre-selection, yet it is not described in the methodology. This issue should be thoroughly explained.

2. A list of definitions of clinical entities used in this study should be explained. What the Authors regarded as: the acute stroke, TIA, ischemic stroke, early stroke recurrence, etc.

3. How the internal carotid arteries were evaluated in detail (which sonographic set and probes, who was performing the examinations, definition of stenosis, were other features of the plaque /e.g. signs of plaque vulnerability/ assessed, etc.

4. The Authors should discuss their finding in the context of other studies on ABCD scale. There are many such papers and they should be citied and discussed.

5. The paper is written in poor English; some parts of the text are ambiguous. The text needs proofreading.

Comments on the Quality of English Language

he paper is written in poor English; some parts of the text are ambiguous. The text needs proofreading.

Author Response

We thank the reviewers for the constructive comments. We have made revisions to the manuscript to address the questions and comments raised by the reviewers. We have highlighted changes made to the original version by setting the text color to red. Our specific responses to each comment are as follows:

Response to Reviewer: 1

Comments to the Author

Although it is potentially a valuable paper, there are several problems that should be addressed before proper review.

  • In general population a majority of strokes are of cardiac origin (usually due to atrial fibrillation-associated embolism; here all patients presented with stenoses of their carotid/cerebral arteries. Probably there was a pre-selection, yet it is not described in the methodology. This issue should be thoroughly explained.
  • Thank you for the comment. In our study, there was no pre-selection for the enrolled TIA patients. Our study was conducted in a teaching hospital, a single center of the level 1 advanced emergency response hospital. The prevalence of the subtypes of acute ischemic stroke is similar to the national survey in Asia countries. In these survey, large vessel is the most common subtypes of acute ischemic stroke, followed by small vessel occlusion and embolic infarction. While the large vessel occlusion and small vessel occlusion account for 37.1% and 23.3% of acute ischemic stroke in Korea, for 30.8% and 26.4% in China, and 21% and 27.5% in Taiwan. For embolic infarction or stroke of cardiac origin, the prevalence was approximately 10% in China, and 15-20% in Korea.(1-3) Compared to the national surveys, our prevalence of large and small vessel occlusion, and embolic infarction were comparable. In addition, previous studies showed the prevalence of moderate and high grades of internal carotid stenosis (>50%) in the TIA population was approximately 8.5 to 21.8%. (4-7)  These data should support no pre-selection bias in our TIA population.

  • A list of definitions of clinical entities used in this study should be explained. What the Authors regarded as: the acute stroke, TIA, ischemic stroke, early stroke recurrence, etc.
  • Thank you for the comment. We have consistently described the clinical entities of TIA and acute ischemic stroke to prevent misunderstandings. First, TIA is defined using a time-based approach, with a diagnosis based on unremarkable findings on non-contrast CT and resolution of neurological deficits within 24 hours.(8-10) Second, acute ischemic stroke is defined as an episode of neurological dysfunction caused by focal cerebral infarction, and a compatible evidence of infarction on diffusion-weighted imaging (DWI) sequences of magnetic resonance imaging.(11)

Accordingly,  acute ischemic stroke was classified into large vessel occlusion, small vessel occlusion, embolic infarction, and other subtypes according to the TOAST classification. (11, 12)

  • How the internal carotid arteries were evaluated in detail (which sonographic set and probes, who was performing the examinations, definition of stenosis, were other features of the plaque /e.g. signs of plaque vulnerability/ assessed, etc.
  • Thank you for the comment. In our study, the ultrasound machine used for carotid vessel evaluation was Affiniti 70 with a linear probe (Philips & Co., Netherlands). Carotid duplex scans were performed by certified ultrasound technicians, and the results were reviewed and verified by a physician. Additionally, the method for measurement of carotid stenosis was according to the European Carotid Surgery Trial (ECST) criteria: the percentage of ICA stenosis = (1 - [diameter of the most stenotic part/estimated original diameter at the site of the stenosis]) x 100.(13) Since our study focused on rapid assessment of the ABCD2 score as well as ICA stenosis percentage in an emergency setting, we did not record plaque vulnerability in this study. We have mentioned this limitation in our discussion section.

  • The Authors should discuss their finding in the context of other studies on ABCD scale. There are many such papers and they should be citied and discussed.
  • Thank you for the comment. We have added more detailed discussion for ABCD2 scale, in association with carotid artery stenosis in our revised manuscript. First, The ABCD2 score(14, 15) stands as the most widely used and simplest risk stratification tool for TIA patients. However, a notable limitation of the ABCD2 score is its failure to incorporate the extent of carotid artery stenosis. Second, in 2014, the ABCD3-I score or ABCD3-I (d,c/i) score(6, 16) was introduced, encompassing additional items such as dual TIA, neuroimaging revealing old or new infarction on DWI sequence of MRI, and ipsilateral carotid artery stenosis or major cerebral artery >50%. Third, while the ABCD3-I score offers a more comprehensive assessment, it also adds complexity, expanding the total points range from the 0–7 to 0–13. Addressing this concern, we proposed a simplified modification of ABCD2-I score by solely integrating neuroimaging of Doppler ultrasound to evaluate carotid artery stenosis. Fourth, the ABCD3 score(7) includes an item called "dual TIA in preceding 7 days," which we did not add to our models for several reasons. Our study tracked patients for three days post-visit, making "dual TIA" unsuitable. Lastly, ABCD3 score was developed in a secondary care setting and designed for a longer follow-up time (30-90 days).(7, 16, 17) Many TIA patients are older and may struggle to describe their symptoms accurately,(18) complicating the inclusion of "dual TIA". Our goal was to create a simple, practical scoring model for the ED. Therefore, ABCD2 or ABCD2-I are more suitable for ED settings.

  • The paper is written in poor English; some parts of the text are ambiguous. The text needs proofreading.
  • We have consulted an English editing service to address any grammatical, spacing, and punctuation errors in the manuscript. All changes made to the original version have been highlighted by setting the text color to red.

  1. Tian D, Yang Q, Dong Q, Li N, Yan B, Fan D. Trends in stroke subtypes and vascular risk factors in a stroke center in China over 10 years. Sci Rep. 2018;8(1):5037.
  2. Jung KH, Lee SH, Kim BJ, Yu KH, Hong KS, Lee BC, et al. Secular trends in ischemic stroke characteristics in a rapidly developed country: results from the Korean Stroke Registry Study (secular trends in Korean stroke). Circ Cardiovasc Qual Outcomes. 2012;5(3):327-34.
  3. Tsai CF, Sudlow CLM, Anderson N, Jeng JS. Variations of risk factors for ischemic stroke and its subtypes in Chinese patients in Taiwan. Sci Rep. 2021;11(1):9700.
  4. Song B, Fang H, Zhao L, Gao Y, Tan S, Lu J, et al. Validation of the ABCD3-I score to predict stroke risk after transient ischemic attack. Stroke. 2013;44(5):1244-8.
  5. Ildstad F, Ellekjaer H, Wethal T, Lydersen S, Fjaertoft H, Indredavik B. ABCD3-I and ABCD2 Scores in a TIA Population with Low Stroke Risk. Stroke Res Treat. 2021;2021:8845898.
  6. Kiyohara T, Kamouchi M, Kumai Y, Ninomiya T, Hata J, Yoshimura S, et al. ABCD3 and ABCD3-I scores are superior to ABCD2 score in the prediction of short- and long-term risks of stroke after transient ischemic attack. Stroke. 2014;45(2):418-25.
  7. Merwick A, Albers GW, Amarenco P, Arsava EM, Ay H, Calvet D, et al. Addition of brain and carotid imaging to the ABCD(2) score to identify patients at early risk of stroke after transient ischaemic attack: a multicentre observational study. Lancet Neurol. 2010;9(11):1060-9.
  8. Douglas VC, Johnston CM, Elkins J, Sidney S, Gress DR, Johnston SC. Head computed tomography findings predict short-term stroke risk after transient ischemic attack. Stroke. 2003;34(12):2894-8.
  9. Marshall J. The Natural History of Transient Ischaemic Cerebro-Vascular Attacks. Q J Med. 1964;33:309-24.
  10. A classification and outline of cerebrovascular diseases. II. Stroke. 1975;6(5):564-616.
  11. Correction to: An Updated Definition of Stroke for the 21st Century: A Statement for Healthcare Professionals From the American Heart Association/American Stroke Association. Stroke. 2019;50(8):e239.
  12. Adams HP, Jr., Bendixen BH, Kappelle LJ, Biller J, Love BB, Gordon DL, et al. Classification of subtype of acute ischemic stroke. Definitions for use in a multicenter clinical trial. TOAST. Trial of Org 10172 in Acute Stroke Treatment. Stroke. 1993;24(1):35-41.
  13. Warlow CP. Symptomatic patients: the European Carotid Surgery Trial (ECST). J Mal Vasc. 1993;18(3):198-201.
  14. Johnston SC, Rothwell PM, Nguyen-Huynh MN, Giles MF, Elkins JS, Bernstein AL, et al. Validation and refinement of scores to predict very early stroke risk after transient ischaemic attack. Lancet. 2007;369(9558):283-92.
  15. Rothwell PM, Giles MF, Flossmann E, Lovelock CE, Redgrave JN, Warlow CP, et al. A simple score (ABCD) to identify individuals at high early risk of stroke after transient ischaemic attack. Lancet. 2005;366(9479):29-36.
  16. Knoflach M, Lang W, Seyfang L, Fertl E, Oberndorfer S, Daniel G, et al. Predictive value of ABCD2 and ABCD3-I scores in TIA and minor stroke in the stroke unit setting. Neurology. 2016;87(9):861-9.
  17. . !!! INVALID CITATION !!! {}.
  18. Bots ML, van der Wilk EC, Koudstaal PJ, Hofman A, Grobbee DE. Transient neurological attacks in the general population. Prevalence, risk factors, and clinical relevance. Stroke. 1997;28(4):768-73.

Reviewer 2 Report

Comments and Suggestions for Authors

Being able to predict recurrent transient symptoms or stroke in patients with TIA would be helpful.  However I find several confusing points in your paper.

-why do you think that stenosis of >50% is more predictive than stenosis of >70%? This is counterintuitive and is different from the results of the NASCET study.

-the number of patients in your study who developed recurrent symptoms in 3 days is extremely high (39%). This is higher than any values I have seen in the literature.  Based on your data any patient with symptomatic >50% stenosis should undergo emergency intervention.

-I have trouble determining at times whether you are describing strokes or TIA

Comments on the Quality of English Language

Needs improvement.

Author Response

We thank the reviewers for the constructive comments. We have made revisions to the manuscript to address the questions and comments raised by the reviewers. We have highlighted changes made to the original version by setting the text color to red. Our specific responses to each comment are as follows:

Response to Reviewer: 1

Comments to the Author

Being able to predict recurrent transient symptoms or stroke in patients with TIA would be helpful.  However I find several confusing points in your paper.

  • why do you think that stenosis of >50% is more predictive than stenosis of >70%? This is counterintuitive and is different from the results of the NASCET study.
  • Thank you for the comment. First, our study should not contradict the results of the NASCET study(1), which had pre-selection criteria for a high grade carotid stenosis. The NASCET study was terminated early for an obvious benefit of endarterectomy in high-grade carotid stenosis. In our study, the prevalence of moderate- and high-grade carotid stenosis (>50%) was similar to general TIA populations.(2-5)
  • Second, the inclusion criteria in the NASCET study was image-oriented, whereas our study was based on symptoms and signs. Our target population was the general population with TIA. Since the NASCET study and our analysis on TIA population targeted on different scenario, we did not enroll adequate sample size of very high grade of CCA and ICA stenosis > 70%. In fact, our study only enrolled 4 patients with CCA stenosis >70% and 1 patient with ICA stenosis >70%, respectively. Consequently, the ABCD2-I (c70) could not discriminate the TIA patients at high risk of subsequent acute ischemic stroke in our study. We have addressed this limitation in the discussion.
  • The detail description of our population characteristics is listed in Table 2.

Characteristic Variables

Total (N = 151)

Revisit
(with acute ischemic stroke)

(N=60)

Discharge
(without acute ischemic stroke)

(N=90)

CCA stenosis > 30%

37.1% (56/151)

48.3% (29/60)

29.7% (27/91)

CCA stenosis > 50%

17.9% (27/151)

25.0% (15/60)

13.2% (12/91)

CCA stenosis > 70%

2.7% (4/151)

2.2% (2/60)

3.3% (2/91)

ICA stenosis > 30%

14.6% (22/151)

23.3% (14/60)

8.8% (8/91)

ICA stenosis > 50%

6.6% (10/151)

24.2% (8/60)

2.2% (2/91)

ICA stenosis > 70%

0.7% (1/151)

1.7% (1/60)

0% (0/91)

  • the number of patients in your study who developed recurrent symptoms in 3 days is extremely high (39%). This is higher than any values I have seen in the literature. Based on your data any patient with symptomatic >50% stenosis should undergo emergency intervention.
  • Thank you for the comment. In our literature review, while excluding patients with the low- and moderate ABCD and the ABCD2 models, we found similar prevalence of developed acute ischemic stroke in high-ABCD2 score population in 2 days, such as 46% in the USA,(6) 33% in Spain,(7) 47% in Italy,(8) and 33% in Canada(9).  Additionally, studies demonstrated high prevalence of subsequent acute ischemic stroke in 7 days for the  high-risk TIA patients, namely the ABCD or ABCD2 scores >3, in a range for the prevalence from 29% to 100%.(10) In our study, there were approximately 75% of the population had an ABCD2 score of >3. We speculated this might explain the higher acute ischemic stroke rate within 3 days in our results.
  • However, this was a single center study, which may limit the generalizability of the findings. We have added this limitation to our article.

  • I have trouble determining at times whether you are describing strokes or TIA
  • Thank you for the comment. We have consistently described the clinical entities of TIA and acute ischemic stroke to prevent misunderstandings. First, TIA is defined using a time-based approach, with a diagnosis based on unremarkable findings on non-contrast CT and resolution of neurological deficits within 24 hours.(11-13) Second, acute ischemic stroke is defined as an episode of neurological dysfunction caused by focal cerebral infarction, and a compatible evidence of infarction on diffusion-weighted imaging (DWI) sequences of magnetic resonance imaging.(14)

Accordingly,  acute ischemic stroke was classified into large vessel occlusion, small vessel occlusion, embolic infarction, and other subtypes according to the TOAST classification. (14, 15)

References

  1. North American Symptomatic Carotid Endarterectomy Trial C, Barnett HJM, Taylor DW, Haynes RB, Sackett DL, Peerless SJ, et al. Beneficial effect of carotid endarterectomy in symptomatic patients with high-grade carotid stenosis. N Engl J Med. 1991;325(7):445-53.
  2. Kiyohara T, Kamouchi M, Kumai Y, Ninomiya T, Hata J, Yoshimura S, et al. ABCD3 and ABCD3-I scores are superior to ABCD2 score in the prediction of short- and long-term risks of stroke after transient ischemic attack. Stroke. 2014;45(2):418-25.
  3. Ildstad F, Ellekjaer H, Wethal T, Lydersen S, Fjaertoft H, Indredavik B. ABCD3-I and ABCD2 Scores in a TIA Population with Low Stroke Risk. Stroke Res Treat. 2021;2021:8845898.
  4. Merwick A, Albers GW, Amarenco P, Arsava EM, Ay H, Calvet D, et al. Addition of brain and carotid imaging to the ABCD(2) score to identify patients at early risk of stroke after transient ischaemic attack: a multicentre observational study. Lancet Neurol. 2010;9(11):1060-9.
  5. Song B, Fang H, Zhao L, Gao Y, Tan S, Lu J, et al. Validation of the ABCD3-I score to predict stroke risk after transient ischemic attack. Stroke. 2013;44(5):1244-8.
  6. Johnston SC, Gress DR, Browner WS, Sidney S. Short-term prognosis after emergency department diagnosis of TIA. JAMA. 2000;284(22):2901-6.
  7. Purroy F, Molina CA, Montaner J, Alvarez-Sabin J. Absence of usefulness of ABCD score in the early risk of stroke of transient ischemic attack patients. Stroke. 2007;38(3):855-6; author reply 7.
  8. Sciolla R, Melis F, Group S. Rapid identification of high-risk transient ischemic attacks: prospective validation of the ABCD score. Stroke. 2008;39(2):297-302.
  9. Coutts SB, Simon JE, Eliasziw M, Sohn CH, Hill MD, Barber PA, et al. Triaging transient ischemic attack and minor stroke patients using acute magnetic resonance imaging. Ann Neurol. 2005;57(6):848-54.
  10. Bhatt A, Jani V. The ABCD and ABCD2 Scores and the Risk of Stroke following a TIA: A Narrative Review. ISRN Neurol. 2011;2011:518621.
  11. Douglas VC, Johnston CM, Elkins J, Sidney S, Gress DR, Johnston SC. Head computed tomography findings predict short-term stroke risk after transient ischemic attack. Stroke. 2003;34(12):2894-8.
  12. Marshall J. The Natural History of Transient Ischaemic Cerebro-Vascular Attacks. Q J Med. 1964;33:309-24.
  13. A classification and outline of cerebrovascular diseases. II. Stroke. 1975;6(5):564-616.
  14. Correction to: An Updated Definition of Stroke for the 21st Century: A Statement for Healthcare Professionals From the American Heart Association/American Stroke Association. Stroke. 2019;50(8):e239.
  15. Adams HP, Jr., Bendixen BH, Kappelle LJ, Biller J, Love BB, Gordon DL, et al. Classification of subtype of acute ischemic stroke. Definitions for use in a multicenter clinical trial. TOAST. Trial of Org 10172 in Acute Stroke Treatment. Stroke. 1993;24(1):35-41.

Reviewer 3 Report

Comments and Suggestions for Authors

The present manuscript titled “ABCD2-I Score Predicts Unplanned Emergency Department  Revisits Within 72 Hours Due to Recurrent Acute Ischemic  Stroke”  by Wei-Zhen Lu et al that ABCD2-I score is useful for predicting the recurrent TIAs within 72 hours of primary TIAs.  The present form of the manuscript is novel and suitable for publication after addressing the following issues.

1.     The present study shows that the ABCD2-I score predicts the reoccurrence of TIAs within 72 hours. The present work finds the risk of secondary attack followed by primary TIA. Authors need to discuss how these findings would help in the treatment of secondary TIAs.

2.     The present study shows that TIAs due to large vessel occlusion can be predicted using the ABCD2-I score. What about the prediction of TIAs with small vessel occlusion? Is this scoring recommended?

3.     Can we predict the reasons for the reoccurrence of TIAs using this scale? What could be done to not get secondary strokes?

4.     This investigation indicates ABCD2 and 2-I score identifies the moderate risk of TIAs. As to the results, hypertension and diabetes are the reasons for these high scores. I am curious to know whether patients with these comorbidities are treated or not. If treated, are the treatment criteria considered for this study?

5.     This study finds 67.9% (36/60) revisited with recurrent stroke whereas 74.7% (68/91) were discharged without stroke (See Table 4). Based on this scoring (ABCD2-I (C50), It seems that highest number of stroke patients discharged without stroke than the patients revisited with stroke. This raises a concern about the validity of the ABCD2-I (C50) scoring.

6.     The are a few grammatical, space, and punctuation errors in the manuscript. Correct them.

Comments on the Quality of English Language

The are a few grammatical, space, and punctuation errors in the manuscript. Correct them.

Author Response

We thank the reviewers for the constructive comments. We have made revisions to the manuscript to address the questions and comments raised by the reviewers. We have highlighted changes made to the original version by setting the text color to red. Our specific responses to each comment are as follows:

Response to Reviewer: 1

Comments to the Author

The present manuscript titled “ABCD2-I Score Predicts Unplanned Emergency Department Revisits Within 72 Hours Due to Recurrent Acute Ischemic  Stroke”  by Wei-Zhen Lu et al that ABCD2-I score is useful for predicting the recurrent TIAs within 72 hours of primary TIAs.  The present form of the manuscript is novel and suitable for publication after addressing the following issues.

  • The present study shows that the ABCD2-I score predicts the reoccurrence of TIAs within 72 hours. The present work finds the risk of secondary attack followed by primary TIA. Authors need to discuss how these findings would help in the treatment of secondary TIAs.
  • Thank you for your insightful comment. Among acute ischemic stroke types, large vessel occlusion accounts for the highest proportion of secondary stroke, followed by small vessel occlusion and embolic infarction.[1-3] Our proposal of ABCD2-I score aims to provide a rapid assessment tool, incorporating the ABCD2 score and sonographic findings in emergency department settings. Compared to the conventional ABCD2 score, our analysis showed that the ABCD2-I (c50) score can more effectively detect medium- and high-risk TIA patients. Consequently, we can easily identify TIA patients with a high risk of acute ischemic stroke within 3 days.
  • In our revision, we have recategorized the patients with ABCD2-I (c50) scores of 0-4 into the low-risk group, those with scores of 5-7 into the moderate-risk group, and those with scores of 8-9 into the high-risk group. Although our study could not determine which specific interventions improve functional outcome, we believed that patients with medium- and high-risk ABCD2-I (c50) scores required close monitoring and management. Regarding interventions to prevent secondary strokes, we considered dual antiplatelet therapy as a potential solution. The Clopidogrel with Aspirin in Acute Minor Stroke or Transient Ischemic Attack (CHANCE) trial reported dual antiplatelet therapy with clopidogrel and aspirin was more effective than aspirin alone for TIA patients in preventing secondary stroke in high-risk TIA patients over 90 days.[4] This evidence could support the use of dual antiplatelet therapy for patients identified as high risk by the ABCD2-I score.

  • The present study shows that TIAs due to large vessel occlusion can be predicted using the ABCD2-I score. What about the prediction of TIAs with small vessel occlusion? Is this scoring recommended?
  • Thank you for your comment. First, the invention of the ABCD2 score aimed to detect high-risk TIAs without targeting specific types of acute ischemic stroke. Both large and small vessel occlusions share common risk factors, such as hypertension and diabetes mellitus.[5] Medium- and high-risk ABCD2 scores are regarded as being associated with extra- and intra-cranial large artery stenosis.[6] Additionally, most large population-based studies demonstrate that recurrence is highest in TIA patients with large vessel occlusion, followed by those with small vessel occlusion.[1-3]
  • Second, while our addition of sonographic findings to the ABCD2 score enhances the identification of asymptomatic large-vessel occlusion, this does not imply that the scoring system is inappropriate for small vessel occlusion. In fact, our cohort included 30% of patients with small vessel occlusion who revisited. However, a larger sample size is needed to confirm the diagnostic performance of the ABCD2-I score specifically for small vessel occlusion.

  • Can we predict the reasons for the reoccurrence of TIAs using this scale? What could be done to not get secondary strokes?
  • Thank you for your comment. First, in our study, both groups exhibited similar ABCD2 scores (mean: 4.60±1.26 vs. 4.35±1.39, p=0.2665). Carotid sonographic findings showed ICA stenosis >30% (23.3% vs 8.8%, p=0.0132) and ICA stenosis >50% (24.2% vs 2.2%, p= 0.0132) were significantly associated with revisit due to acute ischemic stroke. Among the revisit group, 46.7% had large vessel occlusion, followed by 30.0% with small vessel occlusion, and 3.3% with embolic infarction. Consequently, large vessel occlusions significantly contribute to recurrence of TIAs. Previous studies have demonstrated that large vessel occlusions are associated with a higher incidence of early stroke recurrence compared to other types.[1-3]
  • Second, the CHANCE trial demonstrated that dual antiplatelet with clopidogrel and aspirin decreased the 90-day[4] and 1-year risk[7] of acute ischemic stroke in TIA patients with an ABCD2 of ≥ Our ABCD2-I (c30) and ABCD2-I (c50) scores can help detect more TIA patients at high risk of acute ischemic stroke, with the ABCD2-I (c50) showing better discrimination. We believe that TIA patients with the moderate or high risks on the ABCD2-I (c50) score should be monitored more closely. Currently, the National Health Insurance system in Taiwan dose not cover the costs of dual antiplatelet treatment for TIA. Further study is needed to confirm whether dual antiplatelet therapy can improve outcome in these patients. We have added these points in our discussion and limitation section of our manuscript.

  • This investigation indicates ABCD2 and 2-I score identifies the moderate risk of TIAs. As to the results, hypertension and diabetes are the reasons for these high scores. I am curious to know whether patients with these comorbidities are treated or not. If treated, are the treatment criteria considered for this study?
  • Thank you for the comment. First, all of the patients with hypertension and diabetes were treated according to the guidelines during the study period 2014 to 2019. For hypertension, we adhered to the 2015 Guidelines of the Taiwan Society Cardiology and the Taiwanese Hypertension Society for the management of hypertension[8]. For patients with a history of TIA or acute ischemic stroke, antihypertensive medications were prescribed if blood pressure exceeded 220/120 mmHg during the initial 24 hours of the acute stage. Thereafter, blood pressure management aimed for a target of <140/90 mmHg. Since there is no evidence favoring one class of antihypertensive medications over the another, the guidelines focused on achieving the blood pressure goal rather than specifying the types of medications.
  • Second, for diabetes mellitus, we followed the Taiwan Guidelines for the general management of patients with acute ischemic stroke[9] and relevant studies[10,11]. Blood glucose was controlled within the range of 140 to 180 mg/dl, and oral hypoglycemic agents were prescribed as needed. Lastly, a previous suevey on stroke care in Taiwan indicated that the prevalence of hypertension and diabetes mellitus in acute ischemic stroke patients was 79.2% and 45.4%, respectively.[12] In our study, the prevalence of hypertension and diabetes mellitus in TIA patients were comparable to the national survey in Taiwan.

  • This study finds 67.9% (36/60) revisited with recurrent stroke whereas 74.7% (68/91) were discharged without stroke (See Table 4). Based on this scoring (ABCD2-I (C50), It seems that highest number of stroke patients discharged without stroke than the patients revisited with stroke. This raises a concern about the validity of the ABCD2-I (C50) scoring.
  • Thank you for the comment. In the previous version of our manuscript, patients with ABCD2-I (C50) scores of 4-7 were categorized as the moderate-risk group. In this group, 67.9% (36/60) revisited with recurrent stroke, while 74.7% (68/91) were discharged without stroke.
  • In our revised manuscript, we have recategorized the scoring thresholds: patients with ABCD2-I (c50) scores of 5-7 are now considered the moderate risk group, and those with scores of 0-4 are in the low-risk group. Patients with ABCD2-I (c50) scores of 8-9 remain in the high-risk group.
  • Based on the revised recategorization, we found that in the moderate-risk group (ABCD2-I (c50) scores of 5-7), 50.0% (30/60) revisited with acute ischemic stroke and 48.4% (44/91) were discharged without stroke. In the high-risk group (ABCD2-I (c50) scoring system, 11.7% (7/60) revisited with acute ischemic stroke and 2.2% (2/91) were discharged without stroke.
  • These findings indicate that patients with ABCD2-I (c50) score ≥ 5 should be closely monitored to prevent from the occurrence of acute ischemic stroke. We have updated the data presented in Table 4 and 5, as well as Figures 2 and 3, in the latest version of our manuscript to reflect these changes.

Table 4. Predicting Scores of Transient Ischemic Attack Patients with and without Emergency Department Revisit Within 72 h

(N = 151)

Total

Revisit (with acute ischemic stroke)

Discharge (without acute ischemic stroke)

P value

ABCD2-I (c50) score, mean

4.79 ± 1.59

5.13 ± 1.60

4.57 ± 1.56

.0335*

 ABCD2-I (c50) score, distribution

0.0397*

 0-4: Low risk (%)

45.0% (68/151)

38.3% (23/60)

49.5% (45/91)

 5-7: Moderate risk (%)

49.0% (74/151)

50.0% (30/60)

48.4% (44/91)

 8-9: High risk (%)

6.0% (9/151)

11.7% (7/60)

2.2% (2/91)

  • The are a few grammatical, space, and punctuation errors in the manuscript. Correct them.
  • We have consulted an English editing service to address any grammatical, spacing, and punctuation errors in the manuscript. All changes made to the original version have been highlighted by setting the text color to red.

References

  1. Eliasziw, M.; Kennedy, J.; Hill, M.D.; Buchan, A.M.; Barnett, H.J.; North American Symptomatic Carotid Endarterectomy Trial, G. Early risk of stroke after a transient ischemic attack in patients with internal carotid artery disease. CMAJ 2004, 170, 1105-1109, doi:10.1503/cmaj.1030460.
  2. Coutts, S.B.; Eliasziw, M.; Hill, M.D.; Scott, J.N.; Subramaniam, S.; Buchan, A.M.; Demchuk, A.M.; group, V.s. An improved scoring system for identifying patients at high early risk of stroke and functional impairment after an acute transient ischemic attack or minor stroke. Int J Stroke 2008, 3, 3-10, doi:10.1111/j.1747-4949.2008.00182.x.
  3. Smith, W.S.; Lev, M.H.; English, J.D.; Camargo, E.C.; Chou, M.; Johnston, S.C.; Gonzalez, G.; Schaefer, P.W.; Dillon, W.P.; Koroshetz, W.J.; et al. Significance of large vessel intracranial occlusion causing acute ischemic stroke and TIA. Stroke 2009, 40, 3834-3840, doi:10.1161/STROKEAHA.109.561787.
  4. Wang, Y.; Zhao, X.; Liu, L.; Wang, D.; Wang, C.; Li, H.; Meng, X.; Cui, L.; Jia, J.; Dong, Q.; et al. Clopidogrel with aspirin in acute minor stroke or transient ischemic attack. N Engl J Med 2013, 369, 11-19, doi:10.1056/NEJMoa1215340.
  5. Xu, W.H. Large artery: an important target for cerebral small vessel diseases. Ann Transl Med 2014, 2, 78, doi:10.3978/j.issn.2305-5839.2014.08.10.
  6. Bhatt, A.; Farooq, M.U.; Safdar, A.; Hejabian, S.; Razak, A.; Hussain, S.I.; Kassab, M.Y.; Majid, A. ABCD2 Score and Large-Artery Atherosclerosis. Neurohospitalist 2012, 2, 12-17, doi:10.1177/1941874411418239.
  7. Wang, Y.; Pan, Y.; Zhao, X.; Li, H.; Wang, D.; Johnston, S.C.; Liu, L.; Meng, X.; Wang, A.; Wang, C.; et al. Clopidogrel With Aspirin in Acute Minor Stroke or Transient Ischemic Attack (CHANCE) Trial: One-Year Outcomes. Circulation 2015, 132, 40-46, doi:10.1161/CIRCULATIONAHA.114.014791.
  8. Chiang, C.E.; Wang, T.D.; Ueng, K.C.; Lin, T.H.; Yeh, H.I.; Chen, C.Y.; Wu, Y.J.; Tsai, W.C.; Chao, T.H.; Chen, C.H.; et al. 2015 guidelines of the Taiwan Society of Cardiology and the Taiwan Hypertension Society for the management of hypertension. J Chin Med Assoc 2015, 78, 1-47, doi:10.1016/j.jcma.2014.11.005.
  9. Chang, Y.J.; Ryu, S.J.; Chen, J.R.; Hu, H.H.; Yip, P.K.; Chiu, T.F.; Society, C.G.o.T.S. [Guidelines for the general management of patients with acute ischemic stroke]. Acta Neurol Taiwan 2008, 17, 275-294.
  10. Ntaios, G.; Egli, M.; Faouzi, M.; Michel, P. J-shaped association between serum glucose and functional outcome in acute ischemic stroke. Stroke 2010, 41, 2366-2370, doi:10.1161/STROKEAHA.110.592170.
  11. Fuentes, B.; Castillo, J.; San José, B.; Leira, R.; Serena, J.; Vivancos, J.; Dávalos, A.; Nuñez, A.G.; Egido, J.; Díez-Tejedor, E.; et al. The prognostic value of capillary glucose levels in acute stroke: the GLycemia in Acute Stroke (GLIAS) study. Stroke 2009, 40, 562-568, doi:10.1161/STROKEAHA.108.519926.
  12. Hsieh, F.I.; Lien, L.M.; Chen, S.T.; Bai, C.H.; Sun, M.C.; Tseng, H.P.; Chen, Y.W.; Chen, C.H.; Jeng, J.S.; Tsai, S.Y.; et al. Get With the Guidelines-Stroke performance indicators: surveillance of stroke care in the Taiwan Stroke Registry: Get With the Guidelines-Stroke in Taiwan. Circulation 2010, 122, 1116-1123, doi:10.1161/CIRCULATIONAHA.110.936526.

Round 2

Reviewer 2 Report

Comments and Suggestions for Authors

You have tried to address some of my concerns.  I still wonder whether you would recommend more rapid intervention in "high risk" patients than is currently recommended in the literature

Comments on the Quality of English Language

None

Author Response

We thank the reviewers for the constructive comments. We have made revisions to the manuscript to address the questions and comments raised by the reviewers. We have highlighted changes made to the original version by setting the text color to red. Our specific responses to each comment are as follows:

Response to Reviewer: 2

Comments to the Author

Being able to predict recurrent transient symptoms or stroke in patients with TIA would be helpful.  However I find several confusing points in your paper.

  • You have tried to address some of my concerns. I still wonder whether you would recommend more rapid intervention in "high risk" patients than is currently recommended in the literature.
  • Thank you for the comment. Large vessel occlusion is known to have the highest rates of early stroke recurrence,(1-3)and our analysis supports this, showing a high prevalence (46.7%) in revisit patients with large vessel occlusion (Table 2). Studies identify carotid artery stenosis as a major cause of TIA.(1, 4) The methods for carotid artery intervention include carotid endarterectomy (CEA) and carotid artery stenting (CAS). According to the “2021 Guideline for the Prevention of Stroke in Patients With Stroke and TIA,” published by the American Stroke Association,(5) patients with a TIA and symptomatic moderate-grade (50–69%) or high-grade (70–99%) extracranial large vessel occlusion, documented on invasive or non-invasive imaging modalities, are advised to be managed with CEA (Class of Recommendation 1). While a CEA is considered over a CAS within 1 week of the index stroke and in patients of ≥70 years old for revascularization (Class of Recommendation 2a), CAS is preferred over CEA for patients with high-grade (70–99%) stenosis who have medical conditions that increases surgical risk, to reduce procedure-related complications (Class of Recommendation 2a). Medication treatment with single or dual antithrombotic therapy are considered over angioplasty for a moderate- or high-grade intracranial large artery stenosis. (Class of Recommendation 3: Harm).
  • Despite not being explicitly documented in the guideline, studies(6-8) recommend that carotid intervention shall be performed within 2 weeks. The Canadian Stroke Best Practice Recommendations(9) state that carotid intervention should be performed as soon as possible if there is moderate- to high-grade of extracranial ICA stenosis. Excessive delay is considered to cause numerous subsequent acute ischemic stroke events.(6) However, the 2017 Carotid Alarm Study(10) found higher complication rates of death and/or any stroke for the symptomatic carotid stenosis within 48 h. Consequently, carotid intervention is considered safe to be performed between 3 to 7 days. In our revised manuscript, we have mentioned the recommendation of rapid carotid intervention can be performed between 3 to 7 days in the discussion section (the 4th paragraph).

References

  1. Eliasziw M, Kennedy J, Hill MD, Buchan AM, Barnett HJ, North American Symptomatic Carotid Endarterectomy Trial G. Early risk of stroke after a transient ischemic attack in patients with internal carotid artery disease. CMAJ. 2004;170(7):1105-9.
  2. Coutts SB, Eliasziw M, Hill MD, Scott JN, Subramaniam S, Buchan AM, et al. An improved scoring system for identifying patients at high early risk of stroke and functional impairment after an acute transient ischemic attack or minor stroke. Int J Stroke. 2008;3(1):3-10.
  3. Smith WS, Lev MH, English JD, Camargo EC, Chou M, Johnston SC, et al. Significance of large vessel intracranial occlusion causing acute ischemic stroke and TIA. Stroke. 2009;40(12):3834-40.
  4. Markus H, Cullinane M. Severely impaired cerebrovascular reactivity predicts stroke and TIA risk in patients with carotid artery stenosis and occlusion. Brain. 2001;124(Pt 3):457-67.
  5. Kleindorfer DO, Towfighi A, Chaturvedi S, Cockroft KM, Gutierrez J, Lombardi-Hill D, et al. 2021 Guideline for the Prevention of Stroke in Patients With Stroke and Transient Ischemic Attack: A Guideline From the American Heart Association/American Stroke Association. Stroke. 2021;52(7):e364-e467.
  6. Naylor AR. Delay may reduce procedural risk, but at what price to the patient? Eur J Vasc Endovasc Surg. 2008;35(4):383-91.
  7. Naylor AR, Ricco JB, de Borst GJ, Debus S, de Haro J, Halliday A, et al. Editor's Choice - Management of Atherosclerotic Carotid and Vertebral Artery Disease: 2017 Clinical Practice Guidelines of the European Society for Vascular Surgery (ESVS). Eur J Vasc Endovasc Surg. 2018;55(1):3-81.
  8. Kernan WN, Ovbiagele B, Black HR, Bravata DM, Chimowitz MI, Ezekowitz MD, et al. Guidelines for the prevention of stroke in patients with stroke and transient ischemic attack: a guideline for healthcare professionals from the American Heart Association/American Stroke Association. Stroke. 2014;45(7):2160-236.
  9. Wein T, Lindsay MP, Cote R, Foley N, Berlingieri J, Bhogal S, et al. Canadian stroke best practice recommendations: Secondary prevention of stroke, sixth edition practice guidelines, update 2017. Int J Stroke. 2018;13(4):420-43.
  10. Nordanstig A, Rosengren L, Stromberg S, Osterberg K, Karlsson L, Bergstrom G, et al. Editor's Choice - Very Urgent Carotid Endarterectomy is Associated with an Increased Procedural Risk: The Carotid Alarm Study. Eur J Vasc Endovasc Surg. 2017;54(3):278-86.